# Hundreds of myosin 10s are pushed to the tips of filopodia and could cause traffic jams on actin

**Julia Shangguan[1], Ronald S Rock[2]***

[1]Department of Biochemistry and Molecular Biology, University of Chicago, Chicago, United States; [2]Department of Biochemistry and Molecular Biology, The Institute for Biophysical Dynamics, University of Chicago, Chicago, United States

## eLife assessment

The manuscript proposes an alternative method by SDS-PAGE calibration of Halo-Myo10 signals to quantify myosin molecules in filopodia and discusses different scenarios regarding myosin 10 working models to explain intracellular diffusion and targeting to filopodia. Overall, the paper is elegantly written and the methodology is **valuable** in its descriptive potential as these are key numbers to know to ultimately decipher the cellular mechanism of Myo10 action as well as understand the molecular composition of a Myo10-generated filopodium. The evidence for the conclusions is **compelling**, but there are limitations to this study which should be kept in mind when applying this method to other systems.

**\*For correspondence:**
rrock@uchicago.edu

**Abstract** Myosin 10 (Myo10) is a motor protein known for its role in filopodia formation. Although Myo10-driven filopodial dynamics have been characterized, there is no information about the absolute number of Myo10 molecules during the filopodial lifecycle. To better understand molecular stoichiometries and packing restraints in filopodia, we measured Myo10 abundance in these structures. We combined SDS-PAGE densitometry with epifluorescence microscopy to quantitate HaloTag-labeled Myo10 in U2OS cells. About 6% of total intracellular Myo10 localizes to filopodia, where it enriches at opposite cellular ends. Hundreds of Myo10s are in a typical filopodium, and their distribution across filopodia is log-normal. Some filopodial tips even contain more Myo10 than accessible binding sites on the actin filament bundle. Live-cell movies reveal a dense cluster of over a hundred Myo10 molecules that initiates filopodial elongation. Hundreds of Myo10 molecules continue to accumulate during filopodial growth, but accumulation ceases when retraction begins. Rates of filopodial elongation, second-phase elongation, and retraction are inversely related to Myo10 quantities. Our estimates of Myo10 molecules in filopodia provide insight into the physics of packing Myo10, its cargo, and other filopodia-associated proteins in narrow membrane compartments. Our protocol provides a framework for future work analyzing Myo10 abundance and distribution upon perturbation.

## Introduction

Myosins are a group of motor proteins that travel along the cell's dynamic cytoskeletal highways, binding actin and using ATP as fuel. Apart from myosin 2, the other >40 classes are considered 'unconventional' and often include membrane and cargo-binding tail domains (*Odronitz and Kollmar, 2007*). A myosin of particular interest is myosin 10 (Myo10), a motor protein well known for its role in cellular protrusions (*Kerber and Cheney, 2011*; *Mattila and Lappalainen, 2008*). These protrusions, termed

filopodia, comprise tightly packed, parallel actin filaments and participate in a multitude of processes such as phagocytosis, directed cell migration, growth-cone guidance, and cell-cell adhesion (*Mattila and Lappalainen, 2008*). Myo10 has key implications in health dysregulation (*Courson and Cheney, 2015*). For example, upregulated Myo10 is tied to increased genomic instability (*Mayca Pozo et al., 2021*) and breast cancer aggressiveness (*Cao et al., 2014*).

Myo10's role in filopodia has been widely investigated (*Kerber and Cheney, 2011*; *Courson and Cheney, 2015*; *Zhang et al., 2004*; *Tokuo and Ikebe, 2004*; *Bohil et al., 2006*; *Tokuo et al., 2007*; *Zhu et al., 2007*; *Raines et al., 2012*; *Arjonen et al., 2014*; *He et al., 2017*; *Heimsath et al., 2017*; *Summerbell et al., 2020*; *Miihkinen et al., 2021*). Myo10 expression increases dorsal filopodia, and the cargo-binding tail domains, MyTH4 and FERM, are crucial for filopodia formation (*Bohil et al., 2006*). When Myo10 is overexpressed, it produces many Myo10 tip-localized filopodia (*Bohil et al., 2006*; *Watanabe et al., 2010*; *Berg and Cheney, 2002*). Estimations on Myo10-positive filopodia length, average number of filopodia per area, and the velocities of extension and retraction have been reported (*Bohil et al., 2006*; *Kerber and Cheney, 2011*; *Watanabe et al., 2010*; *Cirilo et al., 2024*; *Petersen et al., 2016*). However, questions regarding the quantity and distribution of Myo10 molecules within the cell still linger. Published images of Myo10 show its prominent localization in the filopodial tip, but a pool of Myo10 in the cell body remains (*Liu et al., 2012*; *Kenchappa et al., 2020*). A quantitative understanding of Myo10 localization could provide further insight into filopodial dynamics, how the crowded environment inside a thin filopodium affects the composition of the filo-podial tip complex, and ultimately how Myo10 dysregulation connects to pathology.

Studies estimating protein numbers in subcellular compartments have been conducted before (*Wu and Pollard, 2005*; *Malla et al., 2022*; *Loiodice et al., 2019*; *Sayyad and Pollard, 2022*). Methods include super-resolution microscopy (*Sayyad and Pollard, 2022*), mass spectrometry (*Zhang et al., 2010*; *Shin et al., 2013*), quantitative western blots (*Wu and Pollard, 2005*), and photobleaching experiments (*Hummert et al., 2021*). We describe here a simple imaging and analysis method exploiting HaloTag labeling technology applied to Myo10 and filopodia. Similar to quantitative western blotting of GFP (*Wu and Pollard, 2005*), our strategy relies on SDS-PAGE densitometry using a fluorescent protein standard to estimate the mean number of Myo10 molecules per cell. In parallel fluorescence microscopy work, we record the fluorescent intensity per cell and convert the per-pixel fluorescent signal to the local number of molecules.

We find that the bulk of Myo10 remains in the cell body, with hundreds of Myo10s in each filopo-dium. Myo10 is unevenly distributed across a cell's filopodia, and some filopodial tips have an excess of Myo10 over accessible actin filament-binding sites. Filopodial initiation from the membrane occurs from a median-sized cluster of 160 Myo10 molecules. Hundreds of Myo10 molecules continue to accumulate in filopodial puncta during filopodial extension phases, but accumulation ceases during filopodial retraction. Nascent filopodial extension, second-phase extension, and retraction rates all diminish with increasing Myo10 amounts, suggesting that a large number of motors suppress filo-podial dynamics. Having the number of Myo10 molecules contextualizes interactions of Myo10 with its cargo, the plasma membrane and actin, building a picture of a densely packed filopodial tip compartment.

## Results

### HaloTag-based cellular protein quantitation for microscopy

We expressed human HaloTag-Myo10 in U2OS cells to visualize its behavior in filopodia. Wildtype U2OS cells produce few filopodia, but exogenous expression of Myo10 induces abundant surface-attached filopodia (*Figure 1A*). This induction suggests that Myo10 is a limiting reagent for filopodia in U2OS cells and allows us to assess the impact of variable Myo10 expression levels. HaloTag labeling was selected due to its high-efficiency binding and the commercial availability of a HaloTag stan-dard protein with a known concentration. Our human, full-length Myo10 construct has an N-terminal HaloTag and C-terminal Flag-tag. Both N-terminal and C-terminal-labeled Myo10 have nearly iden-tical fluorescence staining patterns (*Figure 1—figure supplement 1A*).

To estimate the number of Myo10 molecules within specific cellular compartments, we combined SDS-PAGE and epifluorescence microscopy. First, we loaded lysate from a known number of Myo10-transfected U2OS cells on an SDS-PAGE gel along with known amounts of HaloTag standard protein,

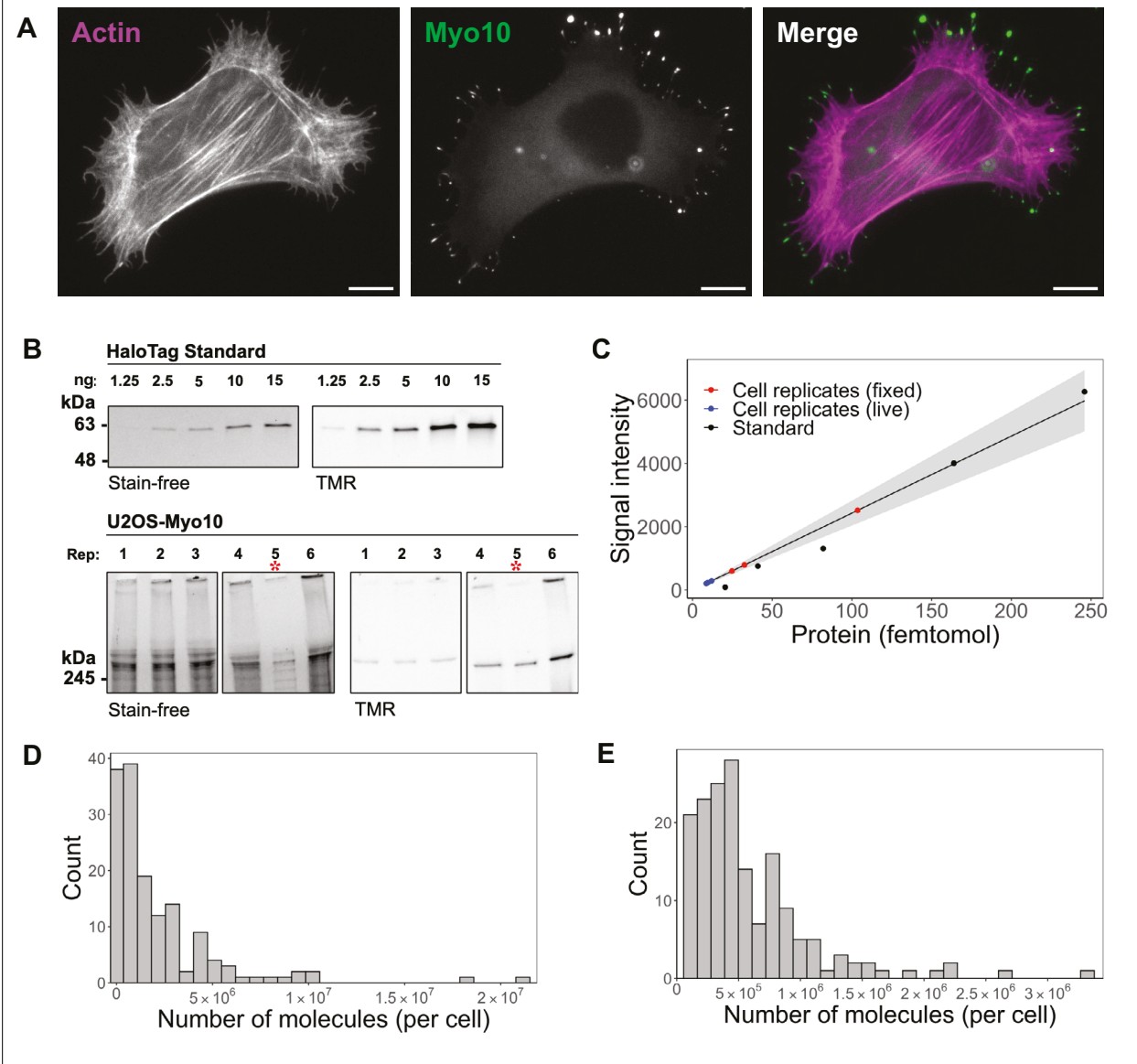

**Figure 1.** Hundreds of thousands of Myo10 monomer molecules found in Myo10-transfected U2OS cells. (**A**) Epifluorescence image of exogenously expressed HaloTag-Myo10 in U2OS cells. Actin is labeled with phalloidin-AF633 (magenta). Myo10 is labeled with HaloTag ligand-TMR (green). Scale bar = 10 µm. (**B**) The top SDS-PAGE lanes show the indicated quantity (in ng) of HaloTag standard protein. The bottom SDS-PAGE lanes from the same gel show 50,000 cells from six separate U2OS transient transfections (except bioreplicate 5, indicated by a red asterisk (*), has 10,000 cells). Bioreplicates 1, 2, and 3 are from live-cell analysis, while bioreplicates 4, 5, and 6 are from fixed-cell analysis. Stain-free shows total protein signal, while TMR shows only TMR-HaloTag-Myo10 signal. Signal was integrated for full-length Myo10 (at ~250 kDa) and any Myo10 aggregated in the wells at the top of the panels. (**C**) Standard curve for TMR fluorescence signal of HaloTag standard protein (black dots) compared to signal from HaloTag-Myo10 U2OS cells (red dots = fixed-cell experiments, blue dots = live-cell experiments). The linear fit is y=24.32x, where the y-intercept is set to 0. $R^2$=0.98. Standard error of slope = 1.42. Gray shading indicates the 95% confidence interval. (**D**) Distribution of the number of Myo10 molecules per fixed cell (N=150 cell images; min = 39,000, 95% CI: 33,000–46,000; median = 1,000,000, 95% CI: 870,000–1,200,000; max = 21,000,000, 95% CI: 18,000,000–25,000,000; bins = 30). (**E**) Distribution of the number of Myo10 molecules per live cell as determined by quantification of the first frame of N=168 cell movies (min = 79,000, 95% CI: 67,000–92,000; median = 450,000, 95% CI: 370,000–520,000; max = 3,300,000, 95% CI: 2,800,000–3,800,000; bins = 30).

The online version of this article includes the following source data and figure supplement(s) for figure 1:

**Source data 1.** Total Myo10 cell signal from all three live-cell bioreplicates.

**Source data 2.** Uncropped and labeled gels for *Figure 1*.

**Source data 3.** Raw unedited gels for *Figure 1*.

**Figure supplement 1.** HaloTag ligand-TMR specifically labels HaloTag-Myo10 and reliably reflects the total Myo10 present in U2OS cells.

*Figure 1 continued on next page*

*Figure 1 continued*

**Figure supplement 1—source data 1.** Uncropped and labeled gels for *Figure 1—figure supplement 1*.

**Figure supplement 1—source data 2.** Raw unedited gels for *Figure 1—figure supplement 1*.

**Figure supplement 2.** HaloTag ligand-TMR robustly labels HaloTag-Myo10.

**Figure supplement 2—source data 1.** Uncropped and labeled gels for *Figure 1—figure supplement 2*.

**Figure supplement 2—source data 2.** Raw unedited gels for *Figure 1—figure supplement 2*.

**Figure supplement 3.** Calculations to convert signal from SDS-PAGE and epifluorescence microscopy to Myo10 molecule counts.

---

a 61 kDa HaloTag-GST fusion protein (*Figure 1B and C*, *Figure 1—figure supplement 1C*). Because all samples were incubated with TMR-HaloTag ligand, we use the standard protein's fluorescence emission in the gel to generate a standard curve and estimate the mean number of Myo10 molecules per cell by densitometry. We propagate the 95% CI from the gel standard curve for our subsequent error estimates. In parallel, we measured TMR-HaloTag Myo10 molecules for the same set of transfected U2OS cells using epifluorescence microscopy. From these measurements, we obtain the total background-subtracted fluorescence intensity for each of the ~50-cell biological replicates (independent transfections on different days) per fixed- and live-cell experiments. We used these totals to determine the fluorescence signal per molecule by microscopy. Both microscopy and gel samples were labeled with excess HaloTag ligand (*Figure 1—figure supplement 2A, D and E*) to ensure maximal labeling of Myo10 molecules with no detectable nonspecific labeling (*Figure 1—figure supplement 1D and E*). All Myo10 molecule measurements accounted for our measured 90% TMR-HaloLigand labeling efficiency (*Figure 1—figure supplement 2B*). Importantly, wildtype U2OS cells do not express detectable levels of Myo10 (*Figure 1—figure supplement 1E*) and generate few filopodia. Therefore, essentially all Myo10 molecules carry the HaloTag label, and these molecules are responsible for inducing the filopodia that we observe. We pooled measurements from the three bioreplicates, resulting in the analysis of 150 cells of varying Myo10 expression levels in fixed-cell experiments, and 168 cells in live-cell experiments (*Figure 1D–E*).

## Limited quantities of Myo10 in filopodia

We start with observations in fixed cells that report the distribution of Myo10 throughout U2OS cells. Exogenous expression of Myo10 results in 12–116 Myo10-positive filopodia per cell (median: 55; *Figure 2A*, *Figure 1—figure supplement 1B*). Varying filopodia density likely reflects cells at different stages of migration or signaling (*Kerber and Cheney, 2011*; *Peuhu et al., 2022*). Despite cells containing a median of 1,000,000 (95% CI: 870,000–1,200,000) total Myo10 molecules (*Figure 1D*), only a small proportion of Myo10 localizes to filopodia (median: 5.4%; *Figure 2B*). This small proportion of filopodial Myo10 is likely due to a limited Myo10 activation signal that is necessary to relieve autoinhibition and begin processive motility along the filopodial shaft.

What does a cell's total Myo10 filopodial signal indicate about filopodia formation? Higher Myo10 filopodial signal correlates with more filopodia, and ~100× more Myo10 appears to boost filopodia by 3× (*Figure 2C*). Likewise, higher total intracellular Myo10 correlates with more filopodia (*Figure 2D*), which supports our earlier hypothesis that Myo10 limits filopodial production in U2OS cells.

## Spatial patterning of Myo10-decorated filopodia

Interactions with membrane-bound proteins, small cytosolic factors, and cortical actin networks might impact where Myo10 is localized. If these influences operate beyond the average separation between filopodia, we might see correlated spatial patterns of higher and lower Myo10 density along the edge of the cell. Indeed, Myo10 is not uniformly distributed, instead concentrating in cellular zones at opposing sides of the cell (*Figure 2E and F*, *Figure 2—figure supplement 1*). Sectors of the cell with the highest quantity of Myo10 have neighboring sectors that are also high (*Figure 2E*, left). Moreover, these high-signal sectors have more Myo10 puncta, as does the opposite side of the cell. Thus, there is a periodic high-low-high-low pattern of punctum density on the cell's perimeter (*Figure 2E*, center; see *Figure 2—figure supplement 2* for an example cell). We define puncta as any cluster/spot of Myo10 detected by segmentation (see Methods for details on image segmentation analysis). However, the number of Myo10 per filopodium is relatively constant on average around the cell (*Figure 2E*, right). Potentially a local activation signal initiates filopodia at a particular site, and then the signal spreads

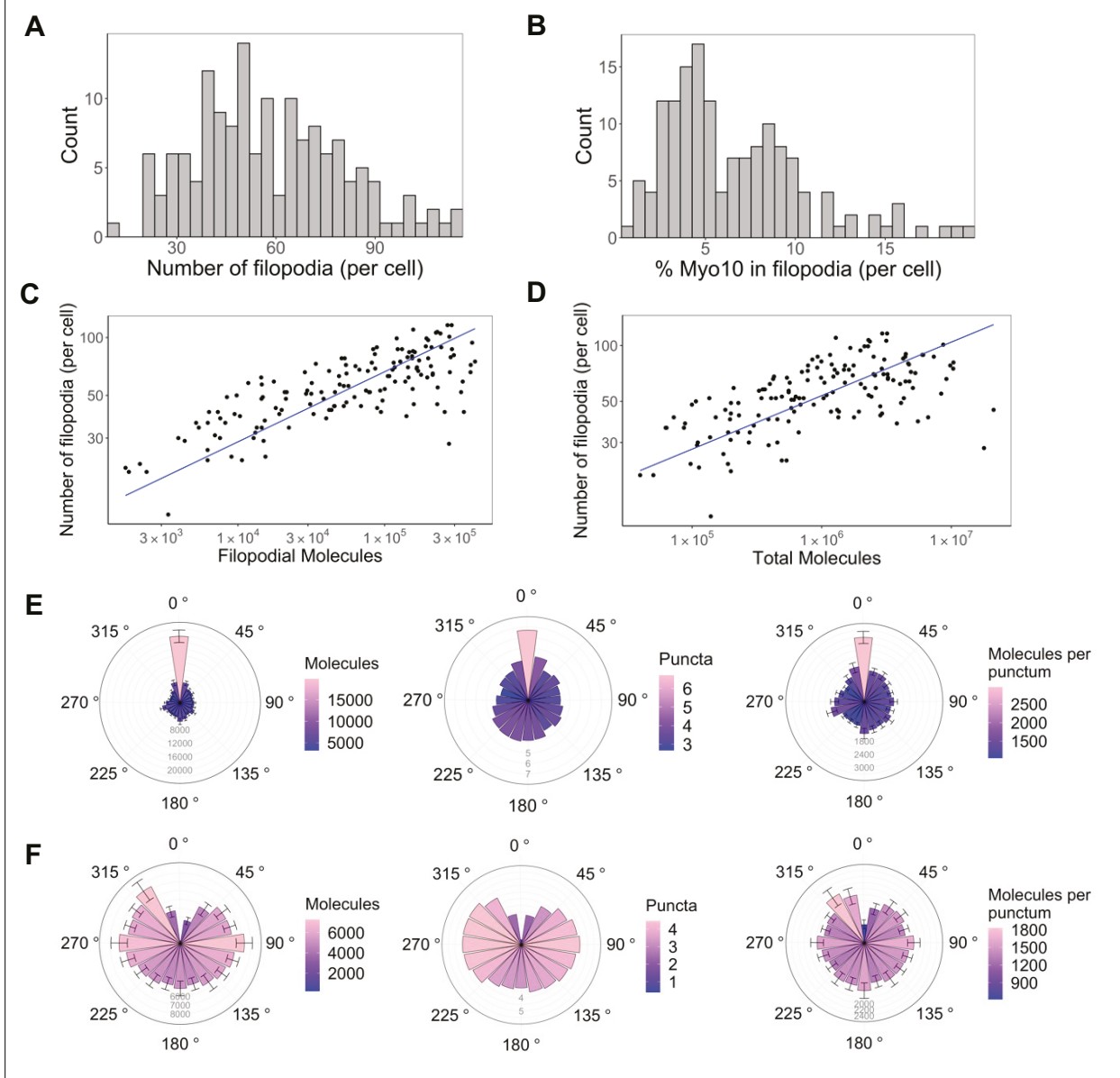

**Figure 2.** Only a small portion of intracellular Myo10 is activated and enters filopodia, and Myo10 is unevenly distributed around the cell. The following values are from fixed-cell images. (**A**) Distribution of the number of Myo10-positive filopodia per cell (N=8733 Myo10-positive filopodia, 150 cells, min = 12, median = 55, max = 116). (**B**) Distribution of the percent of Myo10 localized in the filopodia per cell (N=150 cells, min = 0.86%, median = 5.35%, max = 19.87%). (**C**) Correlation between number of Myo10-positive filopodia in a cell and the filopodial number of Myo10 molecules in the cell. The slope of power law function is 0.36. (**D**) Correlation between number of Myo10-positive filopodia in a cell and the total number of Myo10 molecules in the cell. The slope of power law function is 0.29. (**E**) Spatial correlation of Myo10-rich regions of the cell edge. Each cell was divided into 20 angular sections, and the section with the most molecules was aligned to 0°. Section quantities were then averaged across cells. Molecules, puncta, and molecules per punctum are shown. (**F**) Spatial correlation of Myo10-poor regions of the cell edge. As in E, but the section with the fewest molecules was aligned to 0° for each cell's rose plot. If >1 section contained no Myo10, a randomly selected empty Myo10 section was aligned to 0°. Error bars in E, F are the standard error of the mean for 500 bootstrapped samples of the 150 cells. Note the correlation of both molecules and puncta at opposite ends of cells (0°, 180°), and the anticorrelation with the two sides (90°, 270°).

The online version of this article includes the following source data and figure supplement(s) for figure 2:

**Source data 1.** Filopodial Myo10 signal, cell body Myo10 signal, and number of filopodia for all three fixed-cell bioreplicates.

**Source data 2.** Data for spatial correlation of Myo10-rich regions of the cell edge for all three fixed-cell bioreplicates.

**Source data 3.** Data for spatial correlation of Myo10-poor regions of the cell edge for all three fixed-cell bioreplicates.

**Figure supplement 1.** Myo10 is irregularly distributed across the plasma membrane.

**Figure supplement 2.** Example cell displaying Myo10 density pattern.

from the high Myo10 zone to generate more puncta in the immediate vicinity. Likewise, sectors of the cell with low Myo10 levels tend to be surrounded by fewer Myo10 puncta (*Figure 2F*, center). Some cells even contain membrane regions devoid of filopodia and Myo10 (*Figure 2—figure supplement 1*). We noticed that a filopodial punctum can contain as few as 10 Myo10 molecules (*Figure 3—figure supplement 1A*); these dim Myo10 puncta were primarily tip-localized (*Figure 3—figure supplement 1B and C*).

## Myo10 often saturates accessible actin at filopodial tips

There is a vast range of Myo10 molecules per filopodium (median: 730 molecules, 95% CI: 610–850; *Figure 3A and B*), and its distribution among filopodia is apparently log-normal (*Figure 3—figure supplement 1D and E*). Because log-normal distributions have long tails, we wondered how filopodia can support the movement of extreme Myo10 levels at filopodial tips. To address this question, we compared Myo10 concentrations to the amount of available actin sites in filopodia. We randomly selected a set of 90 filopodial tip-localized Myo10 puncta and measured the apparent length of the punctum. These lengths vary from ~250 nm (diffraction limited) to just over 1.5 µm in the case of the most elongated punctum. To estimate Myo10 concentrations, we used published values to define the geometry of a 'typical' filopodium (*Aramaki et al., 2016*; *Nagy and Rock, 2010*; *Zhuravlev et al., 2012*). Considering a filopodium to be a cylindrical tube, we calculated local concentrations of Myo10 at filopodial tips using the number of molecules, the length of a Myo10 punctum, and a fixed radius of 100 nm (*Mogilner and Rubinstein, 2005*). At the tips, Myo10 ranges from ~6 µM to 560 µM (*Figure 3C and D*).

Interestingly, Myo10 continues to flow into filopodia, even when there may be insufficient actin at the filopodial tip. To estimate the amount of F-actin available for binding, we modeled a filopodium of radius 100 nm comprising 30 actin filaments, of which 16 filaments are on the exposed surface of the bundle. Nagy et al. posited that only 4 of the 13 actin monomers per helical turn are available to Myo10 binding due to steric constraints within a fascin-actin bundle (*Nagy and Rock, 2010*). Using Zhuralev et al.'s equation (*Zhuravlev et al., 2012*) for calculating F-actin monomer concentration in a filopodium, we estimate that ~96 µM of actin monomers are available for Myo10 binding (see *Figure 3—figure supplement 2* for full calculations).

If a filopodium contains ~96 µM of F-actin available to Myo10, and up to 560 µM Myo10, then sometimes Myo10 appears in excess (*Figure 3D and E*). While excess Myo10 is not attached to actin, it may still be docked on the plasma membrane. In this scenario, we estimate available membrane area to include the curved hemisphere area and additional membrane space occupied by the length of the Myo10 tip-localized puncta in *Figure 3C*. We likewise estimate the footprint of the Myo10 C-terminal cargo-binding domains, including those involved in membrane binding (see *Figure 3—figure supplement 3* for details). We find that with even relatively large footprint estimates, there is still sufficient membrane area for filopodial tip-localized Myo10 (*Figure 3F*). In the four cases where the available membrane is exceeded in *Figure 3F*, it is likely because excess Myo10 causes a bulbous extension of the filopodial tip. This extension would provide more area for binding compared to a stereotypical, cylindrical filopodium, since a sphere enclosing a cylinder offers more surface area than a hemispherical cap.

This scenario of excess Myo10 at the filopodial tip would inevitably lead to a molecular traffic jam within the filopodia (*Figure 3G*, *Figure 3—figure supplement 2*), during which Myo10 likely remains bound to the plasma membrane (*Figure 3F*, *Figure 3—figure supplement 3*). Alternatively, the ends of actin bundles could be frayed at filopodial tips, increasing the available actin binding sites for Myo10 (*Figure 3H*).

## Myo10 accumulation over the filopodial lifecycle

All the above observations are based on fixed U2OS cells and therefore provide a static view of filopodial systems. To understand Myo10 levels and how they change during the dynamic filopodial lifecycle, we adapted our fluorescence calibration method for live-cell imaging. We found it critical to rapidly collect 50–60 snapshots of individual cells for the calibration before collecting our movies on each sample. Adequate temperature control of the sample was also crucial, as the U2OS cells would rapidly contract their filopodia in response to cold. We identified and focused on three events in the filopodial lifecycle: filopodial initiation from a Myo10 punctum that appears at the plasma membrane,

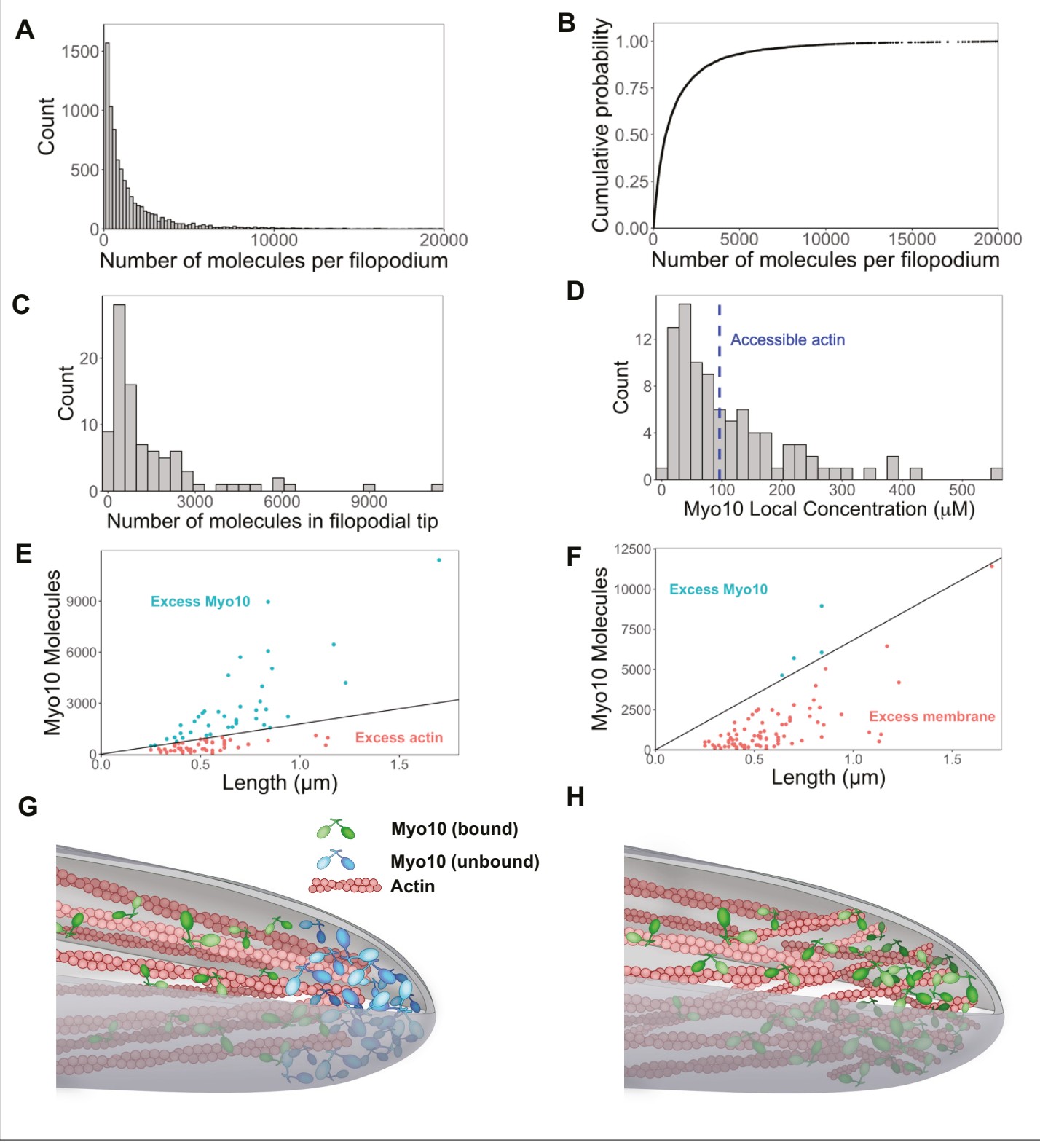

**Figure 3.** Hundreds of Myo10 molecules are found in a filopodium, potentially in excess over available actin at the filopodial tip. The following values are from fixed-cell images. (**A**) Distribution of the number of Myo10 molecules per filopodium (N=150 cells, 8733 filopodia; min = 6, 95% CI: 5–7; median = 730, 95% CI: 610–850; max = 80,000, 95% CI: 67,000–93,000; 62 values >20,000 not shown; bins = 100). (**B**) Cumulative distribution function plot of data in part A. (**C**) Distribution of the number of Myo10 molecules at the filopodial tip (90 randomly chosen filopodia tip-localized Myo10 puncta from nine different cell images; min = 66, 95% CI: 55–76; median = 788, 95% CI: 660–915; max = 11,000, 95% CI: 9600–13,000; bins = 30). (**D**) The local

*Figure 3 continued on next page*

*Figure 3 continued*

concentration of Myo10 at a filopodial tip. To estimate the volume, we measured the length of the filopodia tip-localized Myo10 puncta from part C in ImageJ. We then modeled filopodium as a cylinder of radius = 0.1 µm (published average). Min = 6.2 µM, 95% CI: 5.2–7.2; median = 84 µM, 95% CI: 70–97; max = 560 µM, 95% CI: 470–650, bins = 30. Blue dashed vertical line indicates the concentration of F-actin accessible for Myo10 binding in a filopodium (~96 µM). (**E**) Scatterplots of molecules vs. length for the puncta from part C. The phase boundary shows the 96 µM threshold from part D. (**F**) As in part E, but the line represents an estimate of allowable Myo10 on the filopodial tip membrane area. See *Figure 3—figure supplement 2* for membrane occupancy estimates. (**G**) Model of a Myo10 traffic jam at the filopodium tip. Not enough available actin monomers results in a population of free Myo10 (in blue). The free Myo10 is detached from actin but potentially still membrane-associated. (**H**) Model of frayed actin filaments at the filopodium tip. If actin filaments are not neatly packed into parallel bundles at the filopodium tip, disorganized and frayed actin filaments yield more accessible binding sites to Myo10.

The online version of this article includes the following source data and figure supplement(s) for figure 3:

**Source data 1.** Signal in segmented filopodial Myo10 puncta for all three fixed-cell bioreplicates.

**Source data 2.** Data for local concentration of Myo10 at filopodial tips for all three fixed-cell bioreplicates.

**Figure supplement 1.** HaloTag ligand-TMR labeling to determine Myo10 distributions along filopodia.

**Figure supplement 2.** 96 µM of actin monomers are accessible to Myo10 in a filopodium.

**Figure supplement 3.** Plasma membrane at the filopodial tip can accommodate a portion of Myo10 unbound to actin.

second-phase elongation from a Myo10 tip punctum in an established filopodium (*Watanabe et al., 2010*), and filopodial retraction.

Because Myo10 seems to be limiting filopodial production in U2OS cells, we wondered how many Myo10 molecules are needed for filopodial initiation. As Myo10 coalesces at a site on the plasma membrane and prepares to shoot into a filopodium, a median of 160 molecules gather in the punctum (*Figure 4A*, *Figure 4—video 1*). The minimal value at the initiation site is 52 molecules, which may represent the lower limit of Myo10 needed for filopodial initiation. Although we attempted to observe Myo10 coalescence at the site of nascent filopodium, we were unable to detect them due to the limited sensitivity of the epifluorescence microscope. We did not use our single-molecule TIRF microscopy because it has a limited field of view that is unable to capture an entire U2OS cell, which is needed to perform the fluorescence calibration.

Oftentimes, a new filopodial tip can emerge from an existing one in a process that we call second-phase elongation. Such a Myo10-induced multi-cycle filopodial elongation mechanism has been previously proposed (*He et al., 2017*). We find a median of 290 Myo10 molecules organize and form the separate punctum at the start of second-phase elongation (*Figure 4B*, *Figure 4—video 2*, *Figure 4—figure supplement 1*). This number represents about double the number of Myo10 involved in nascent elongation (median: 160). Retracting Myo10 puncta initially contain a median of 240 molecules (*Figure 4C*).

Both the initial punctum and the second-phase punctum accumulate additional Myo10 molecules over time (*Figure 4D and E*). To estimate the accumulated amount, we took the last point of each trajectory and performed a t-test against the null hypothesis of a zero mean. Initial puncta accumulate a mean of 95 molecules (95% CI: 74, 120; t=8.8; df = 236; p-value = $2.7*10^{-16}$), while second-phase puncta accumulate 307 molecules (95% CI: 130, 480; t=3.5; df = 50; p-value = 0.001). However, unlike these two filopodial growth processes, retracting Myo10 puncta remain relatively constant as they return to the cell body (mean: –73 molecules; 95% CI: −140,–10; t=–2.3; df = 57; p-value = 0.025; *Figure 4F*). When retracting filopodia reach the cell body, the punctum often dissipates. However in a few cases, a new Myo10 punctum will coalesce and emerge from the location of a previous one (*Figure 4—video 3*).

## Increasing quantities of Myo10 slow the punctum movements in all phases of the filopodial lifecycle

Finally, we wondered how the speeds of filopodial elongation and retraction are related to the number of Myo10 molecules in a tip punctum. Such values would help to constrain models that depend on Myo10 flux to deliver components to the filopodial tip or to generate force at the tip to assist actin polymerization. Interestingly, speeds of tip puncta during filopodial initiation, second-phase elongation, and retraction are all inversely related to the amount of Myo10 in the punctum (*Figure 4G–I*). In each case, fewer molecules are associated with a higher variance in speed. The median speed of

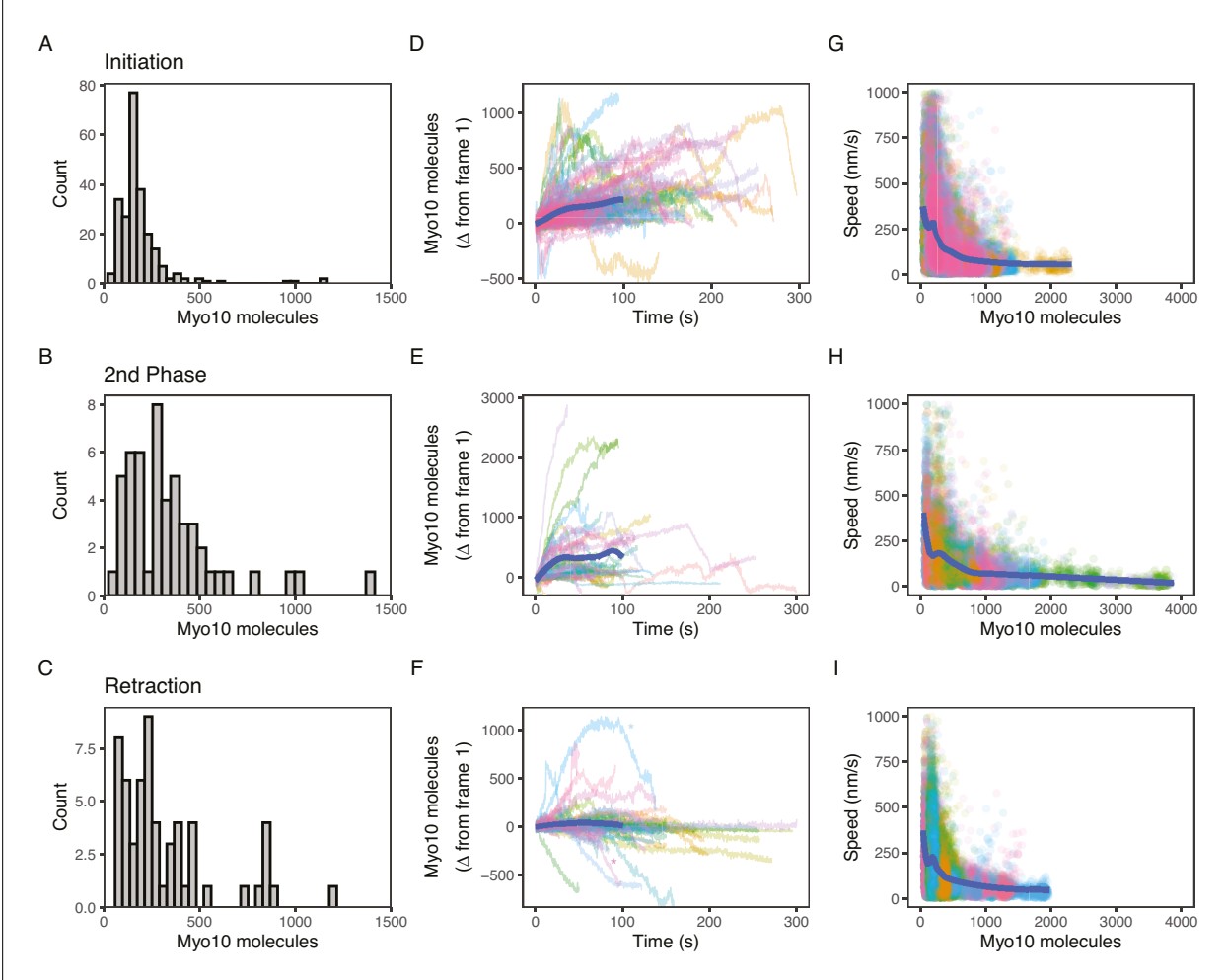

**Figure 4.** Myo10 dynamics from live-cell movies. (**A–C**) Dense Myo10 puncta appear at the start of each phase of the filopodial lifecycle. Distributions of the number of Myo10 molecules in puncta upon: (**A**) Filopodial initiation (min = 52, 95% CI: 44–61; median = 160, 95% CI: 140–190; max = 1200, 95% CI: 970–1300). (**B**) Second-phase elongation (min = 65, 95% CI: 54–75; median = 290, 95% CI: 240–340; max = 1400, 95% CI: 1200–1600). (**C**) Filopodial retraction (min = 71, 95% CI: 60–82; median = 240, 95% CI: 200–280; max = 1200, 95% CI: 1000–1400). Values are the means of the first two frames after spot detection and identification of filopodial lifecycle stage. Histograms A–C have 30 bins each. (**D–F**) Accumulation of Myo10 in puncta after filopodial initiation or second-phase elongation, but not after retraction. Evolution of number of molecules for each filopodial phase over time for: (**D**) filopodial initiation. (**E**) Second-phase elongation. (**F**) Filopodial retraction. Starting values from (**A–C**) were subtracted from all traces to obtain delta over time. The generalized additive model (GAM) trend lines (blue) exclude long times (>100 s) with few surviving trajectories. The outlier trajectory indicated by the magenta asterisk is from *Figure 4—video 3*, and the cyan asterisk is from *Figure 4—video 4*. (**G–I**) Myo10 punctum speeds are inversely correlated with the number of Myo10 molecules. Plots of instantaneous speeds vs. molecules for: (**G**) filopodial initiation (min = 0.7, median = 160, max = 2,200 nm/s, Spearman's $\rho$ = –0.51, p<2.2*10$^{-16}$). (**H**) Second-phase elongation (min speed = 0.4, median = 110, max = 3000 nm/s, Spearman's $\rho$ = –0.55, p<2.2*10$^{-16}$). (**I**) Filopodial retraction (min speed = 1.2, median = 140, max = 1900 nm/s, Spearman's $\rho$ = –0.45, p<2.2*10$^{-16}$). Color signifies Myo10 puncta belonging to the same trajectory within each event type. Colors are independent in each panel. Blue lines are GAM trend lines. Panels A, D, G: 237 trajectories from 31 cells; B, E, H: 51 trajectories from 19 cells; C, F, I: 58 trajectories from 26 cells.

The online version of this article includes the following video, source data, and figure supplement(s) for figure 4:

**Source data 1.** Trajectories of Myo10 puncta upon retraction from live-cell bioreplicate 1.

**Source data 2.** Trajectories of Myo10 puncta upon second-phase elongation from live-cell bioreplicate 1.

**Source data 3.** Trajectories of Myo10 puncta upon filopodial initiation from live-cell bioreplicate 1.

**Source data 4.** Trajectories of Myo10 puncta upon retraction from live-cell bioreplicate 2.

**Source data 5.** Trajectories of Myo10 puncta upon second-phase elongation from live-cell bioreplicate 2.

**Source data 6.** Trajectories of Myo10 puncta upon filopodial initiation from live-cell bioreplicate 2.

**Source data 7.** Trajectories of Myo10 puncta upon retraction from live-cell bioreplicate 3.

*Figure 4 continued on next page*

*Figure 4 continued*

**Source data 8.** Trajectories of Myo10 puncta upon second-phase elongation from live-cell bioreplicate 3.

**Source data 9.** Trajectories of Myo10 puncta upon filopodial initiation from live-cell bioreplicate 3.

**Figure supplement 1.** Myo10 is found along the filopodium shaft and can undergo a second elongation.

**Figure 4—video 1.** Myo10 punctum initiates a nascent filopodium.
https://elifesciences.org/articles/90603/figures#fig4video1

**Figure 4—video 2.** Myo10 punctum can undergo a second elongation.
https://elifesciences.org/articles/90603/figures#fig4video2

**Figure 4—video 3.** Example of Myo10 punctum decreasing in intensity as it returns to cell body.
https://elifesciences.org/articles/90603/figures#fig4video3

**Figure 4—video 4.** Example of Myo10 punctum in a 'snapping' filopodium as it returns to cell body.
https://elifesciences.org/articles/90603/figures#fig4video4

elongating Myo10 puncta is 160 nm/s, which is similar to previously published values (*Nagy et al., 2008*; *Kerber et al., 2009*; *Kerber and Cheney, 2011*; *Watanabe et al., 2010*). Consistent with the trend of more molecules resulting in slower puncta velocities, the larger puncta involved in second-phase elongation display a slower median speed of 110 nm/s. The median speed of retracting Myo10 puncta is 110 nm/s, similar to previously reported values (*Watanabe et al., 2010*). For retracting puncta moving faster than actin retrograde flow, the velocity of Myo10 likely reflects a collapse of the whole filopodium or other complex filopodial dynamics. Our live movies capture some of these non-constant Myo10 retrograde events. *Figure 4—video 4* exemplifies a decreasing intensity trajectory, wherein a Myo10 punctum rapidly 'snaps' closer to the cell body before diffusing into the cytoplasm. The 'snap' could be explained by the filopodium undergoing coiling or buckling, possibly induced in part by Myo10-generated force (*Leijnse et al., 2022*; *Leijnse et al., 2015*).

## Discussion

### Limited activation of Myo10

Myo10 is the limiting reagent for constructing filopodia in U2OS cells, but less than 20% arrives in filopodia when exogenously expressed (*Figure 2B*). There is ample time for Myo10 to diffuse from the cytosolic pool to the numerous filopodial sites present in these cells, where it would then be captured. Thus, we expect that most of the cytosolic pool of Myo10 is in an inactive, autoinhibited state. Available PtdIns(3,4,5)$P_3$ (PIP$_3$) in the cell could be limiting the portion of free intracellular Myo10 that is activated and entering filopodia. Umeki et al. proposed that Myo10 binds PIP$_3$ at cell peripheries and dimerizes, becomes activated, and converges local actin filaments to form the filopodial base (*Umeki et al., 2011*). Although Myo10 can participate in filopodia formation independently of VASP proteins and substrate attachment (*Bohil et al., 2006*), other proteins have been found to be conserved in mechanisms of filopodia initiation. Components such as Cdc42 (*Krugmann et al., 2001*), formin (*Alieva et al., 2019*), Arp2/3 complex (*Korobova and Svitkina, 2008*), neurofascin (*Peuhu et al., 2022*), or VASP (*Lebrand et al., 2004*; *Arthur et al., 2021*) could also be prospective limiting partners of Myo10 entry into filopodia.

Prior information on the absolute number of Myo10 molecules in filopodia is sparse. *Kerber et al., 2009* transfected HeLa cells with low levels of GFP-Myo10 (6–12 hr post-transfection) and found 20–200 molecules at the filopodial tip (). These levels are consistent with the range that we measure here (*Figure 3A*). The shorter time post-transfection clearly affects Myo10 expression levels, which is evident in our data. Fixed-cell experiments had higher Myo10 expression overall (48 hr post-transfection) compared to live-cell experiments (24 hr post-transfection) (*Figure 1D and E*).

### Myo10 accretion into filopodial puncta

The distribution of Myo10 among filopodial puncta is log-normal rather than normal (*Figure 3A and B*, *Figure 3—figure supplement 1*). This stands in contrast to more consistent, normally distributed quantities of other actin-binding proteins. In fission yeast, proteins in the spindle pole body (e.g. Sad1p), at the cytokinetic contractile ring (e.g. Myo2p, Rho GEF Rgf1p), and within actin patches (e.g. ARPC1, capping protein Acp2p) did not display the wide concentration ranges that we saw for Myo10

in filopodia (*Wu and Pollard, 2005*). We propose several reasons for the non-Gaussian distribution of Myo10 molecules. Log-normal distributions are commonly associated with growth processes, and the accumulation of Myo10 at filopodial tips fits this description. Filopodia have a large capacity for additional Myo10, and an ample reserve of unactivated Myo10 remains in the cytosol. Slow activation of Myo10 would then allow its travel down filopodia and accretion into puncta. Filopodia operate far from saturation, enabling relatively unrestricted Myo10 accumulation. In contrast, we suspect that the spindle pole body, cytokinetic ring, and actin patches represent cytoskeletal systems constructed from one or more limiting reagents. Limiting reagents put an upper bound on the growth of the system and lead to normal distributions at saturation.

Furthermore, the end requirements of cytokinesis and filopodial formation differ. Cytokinesis is a tightly regulated process involving a balance of forces, and the precise timing of each stage has been described in fission yeast (*Wu et al., 2003*). Disruptions to components of the contractile ring, such as myosin inhibition, affect rates of actin filament disassembly and ring constriction (*Malla et al., 2022*). In contrast, there is no specific number of filopodia that cells aim to create nor is there an optimal number of Myo10 per filopodium. Therefore, filopodia formation is a much more permissive process than cytokinesis, which could also explain the variation in values we observe of Myo10 compared to contractile ring-associated proteins.

## Non-uniform Myo10 filopodial distribution

Myo10 tends to accumulate in filopodia that are at opposite sides of the cell. These high Myo10 sectors have a higher density of filopodia but a similar quantity of Myo10 per filopodium (*Figure 2E and F*). We hypothesize a few different explanations for why Myo10 and filopodia not equally distributed around the cell. Potentially, Myo10 is funneled into filopodia that are moving toward specific environmental stimuli that simultaneously affect Myo10 (*Meyen et al., 2015*; *Efremov et al., 2022*). For example, local generation of $PIP_3$ might serve to dock and activate Myo10 in focused regions of the plasma membrane, such as the basolateral surfaces of polarized MDCK cells (*Lu et al., 2011*; *Liu et al., 2012*). The docked but diffusing Myo10 is then more likely to initiate and elongate new filopodium. This overall diffusion and capture process for Myo10 has been observed at the single-molecule level in cells, where it has been described as a 3D to 2D to 1D reduction in dimensionality (*Baboolal et al., 2016*).

Alternatively, Myo10 could be responding to information encoded in the actin network itself (e.g. nearby stress fibers) or other local signaling molecules. Myosins can differentiate the structural and chemical properties of actin filaments, including age, tension, curvature, post-translational modifications, and multifilament architecture (*Santos et al., 2020*). Thus, actin filament networks could be partly responsible for choreographing Myo10 trajectories (*Santos et al., 2020*). Future studies could identify potential guidance cues that trigger the initiation of Myo10-positive filopodia.

## An excess of Myo10 at the filopodial tip

A major finding of this work is that there is often an excess of Myo10 at filopodial tips over the available actin-binding sites for myosin (*Figure 3E–H*). Interestingly, there is a mechanism that continues to push Myo10 into the filopodia, even when there may not be available actin to bind. This observation points to two notable features of motility in filopodial systems. The first is that there is no mechanism for a filopodium to shut off entry at the base when the tip is already overfilled with Myo10. The two sites are functionally independent of each other, although retracting filopodia differ in this regard (see below). The second is that Myo10 does not pack in an orderly fashion at the filopodial tip. Although our observations are in the context of an overexpression system, we note that immunofluorescence staining of endogenous Myo10 also shows occasional high-intensity, bulbous filopodial tips (*Berg and Cheney, 2002*). Our concentration estimates at the filopodial tip suggest that there is sufficient plasma membrane area for the Myo10. Therefore, even though Myo10 disengages from actin, it may remain engaged to the plasma membrane through its PH domains (*Figure 3F*, *Figure 3—figure supplement 3*). As Myo10 molecules diffuse out of the concentrated region at the tip, they can encounter open sites on the actin filament bundle and reengage, walking back to the tip.

The structural model of the filopodium discussed above assumes an orderly, hexagonally closely packed bundle structure all the way to the filopodial tip. Consistent with this picture, EM images of mouse melanoma B16F1 cells indicate bundled parallel actin filaments at the filopodial tip (*Svitkina*

*et al., 2003*). However, this is not always the case. In *Dictyostelium discoideum* cells, actin filaments in the filopodia tip can terminate in free ends and/or appear fragmented, by which short actin filaments are arranged into a non-parallel array (*Medalia et al., 2007*). Bent filopodial tips in mammalian cells overexpressing GFP-Myo10 can display bulbous membrane extensions containing splayed actin filaments or actin filaments arranged in loops (*Li et al., 2023*). These fragmented, frayed, or otherwise disorganized ends of the filopodial actin bundle might support more actin-bound Myo10 (*Figure 3H*). Indeed, an excess of Myo10 at the tip would compete with actin crosslinkers and help to support fraying. Further investigation is required to understand how Myo10 traffic jams and actin fraying affect the dynamics of actin itself and the activity of its associated proteins.

## Number of Myo10 molecules needed for filopodial initiation and second-phase elongation

We observe a median of 160 Myo10 molecules and a minimum of 52 molecules at filopodial initiation (*Figure 4A*). These quantities correspond to approximately 5 and 2 molecules per actin filament at a nascent filopodium, respectively. *Watanabe et al., 2010* also saw a rapid (2–20 s) accumulation of Myo10 recruitment at filopodial initiation. Myo10 participates in bridging actin filaments in emergent fascin-actin bundles, the actin structures that it prefers (*Mattila and Lappalainen, 2008*; *Nagy et al., 2008*; *Ropars et al., 2016*). Such bridging could help to organize or orient the bundles to allow protrusion, much like the development of the lambda-precursors observed by *Svitkina et al., 2003*.

Second-phase filopodial elongation involves more molecules of Myo10, a median of 290 (*Figure 4B*). This increase in number likely reflects the larger available pool of active Myo10 present at an established filopodial tip. Because only 160 molecules are needed for filopodial initiation, we believe that Myo10 quantities do not limit second-phase extension if initiation and second-phase elongation have similar mechanical requirements. Focal adhesion proteins, such as vinculin and integrin-β1, have been previously identified as essential components during the second-phase of filopodia extension (*He et al., 2017*). Other actin-associated proteins (e.g. Arp2/3 and vinculin) are also present at this stage (*He et al., 2017*). The duration required for focal adhesion proteins to translocate into a filopodium, coupled with actin crosslinking and bundling in the new filopodium, results in an increased accumulation of Myo10 at the local level. This accumulation is a consequence of Myo10 awaiting interaction with other protein partners at the filopodium tip. We speculate that the entire process takes longer than at the cell body plasma membrane, where the concentration of available focal adhesion proteins is higher.

## Myo10 entry into filopodia switches off upon retraction

Myo10 accumulates within puncta after filopodial initiation and second-phase elongation (*Figure 4D and E*), adding several hundred molecules on the minutes timescale. However, in retracting filopodia, accretion of additional Myo10 ceases (*Figure 4F*). These retracting filopodia frequently have immediate neighbors that are not retracting. Therefore, the mechanism that switches off Myo10 traffic must be highly focused to single filopodia. Moreover, this switching mechanism must operate at the base of the filopodium where the Myo10 traffic would otherwise enter. Two mechanisms seem plausible given these considerations. Either the retracting filopodium generates a Myo10-inhibitory signal that diffuses out and rapidly dissipates, or the actin architecture of the filopodium is altered and blocks Myo10 motility. These findings offer a quantitation of the dynamic behavior of Myo10 during retrograde flow, but further research is necessary to uncover the mechanisms governing switching of filopodial movement.

## Multiple motors interfere to slow dynamic movements at the filopodial tip

The presence of increasing quantities of Myo10 slow filopodial movements in all three phases of the filopodial lifecycle (*Figure 4G–I*). In reconstituted motility systems with multiple molecules, mechanically coupled motors typically move more slowly than uncoupled (or single) motors. This slowing is attributed to strain-sensitive motor coupling and motor dissociation limiting the collective velocity to some degree (*Lu et al., 2012*; *Hariadi et al., 2014*). We propose that similar drag mechanisms operate here to slow the dynamics of filopodia.

Recent studies address the influence of filopodial tip-directed myosin motors on filopodial elongation. In work by *Cirilo et al., 2024*, wildtype and fast-mutant Myo3 motors also localize to filopodial tips, and the filopodial elongation rate correlates with the Myo3 speed. Myo3 is a slower motor than Myo10 (70 nm/s vs. 300–600 nm/s), and the Myo3-decorated filopodia move more slowly than we typically observe here (*Cirilo et al., 2024*). These observations with slower myosins are consistent with our drag-based interference of filopodial dynamics: a coupled collection of slower motors would produce more drag than a collection of faster motors, all else being equal.

## Limitations

Our experiments were conducted with the intention of minimizing biases and errors, but our results do come with caveats. Our results are within the context of an overexpression system, and other systems expressing endogenous Myo10 may express at a lower overall level than here. However, cells expressing endogenous Myo10 also show prominent dense tip localization by immunofluorescence (*Berg and Cheney, 2002*), so the Myo10 organization we see here does exist physiologically. Our goal in this study was to understand Myo10's role in filopodial initiation and dynamics in a defined system where Myo10 is the limiting factor. U2OS cells express undetectable levels of endogenous Myo10 (*Figure 1—figure supplement 1E*), so we are indeed observing the influence of Myo10 in a situation where Myo10 is the limiting factor for filopodia formation.

Regarding aspects of microscopy and automated image processing, very dim Myo10 spots could have escaped detection. During fixed and live imaging, we are unable to detect single molecules of Myo10 because we do not have the sensitivity of TIRF. Furthermore, our estimates of molecules are predicated on the calibration curve of the HaloTag standard protein on the SDS-PAGE gels, which is likely the highest source of error on our molecule counts. Despite these concerns, our values still provide a sense of the magnitude of Myo10 molecules within cellular structures.

## Ideas and speculation

One of our key findings is the excess of Myo10 compared to available actin at filopodial tips. We propose that there is a sizable population of Myo10 docked on the plasma membrane but not to actin (*Figure 4G*). Alternatively, actin filaments that fray from the fascin-actin bundle yield more binding sites for Myo10 (*Figure 4H*). Myo10 moves processively along actin bundles, even disorganized ones formed by crowding reagents (*Nagy et al., 2008*). Any single overloaded filopodium may use either or both mechanisms.

We argue that there is an interplay between incoming Myo10 that release upon encountering a traffic jam of other Myo10 molecules, and Myo10 molecules that are part of the traffic jam that are 'pushed off' (i.e. naturally dissociate and are replaced) by new arrivals. Myo10 cargoes may need to handle either the actin-attached or actin-detached situation, which has implications for force-sensing cargoes such as integrins.

At the filopodial tip, we expect that newly formed actin will be rapidly occupied by Myo10 due to the high quantity on standby. However, fascin must eventually bind to facilitate the bundling of new actin polymerized at the filopodial tip. While Myo10 cannot bind to the interior of a fascin-actin bundle due to insufficient space, the converse is also true: fascin cannot crosslink at a Myo10-bound actin filament site until the Myo10 dissociates. However, this Myo10 dissociation occurs frequently as part of the stepping process. Once there is an opening, fascin can outcompete Myo10 due to the high avidity interaction between fascin and actin. Fascin binding is essentially irreversible, as we saw no disassembly of a suspended fascin-actin bundle even after 1 hr of continuous buffer wash (*Courson and Rock, 2010*). Therefore, the dynamic association and dissociation of Myo10 at the filopodial tip with subsequent, irreversible fascin binding may be essential for the continued elongation of the filopodium.

## Conclusion

Knowing the number of Myo10 molecules in a filopodium provides insight into molecular packing geometries in confined intracellular spaces. This study showcases the high density of molecules that can be packed into tight cellular compartments, captures the number of Myo10 molecules needed for filopodial initiation, and finds that Myo10 entry into filopodia switches off upon filopodial retraction. Our results contribute to understandings of molecular stoichiometries in filopodia and Myo10

abundance in these structures. Our protocol also provides a framework for future studies examining the factors that tune Myo10 density and distribution.

# Methods

## Key resources table

| Reagent type (species) or resource | Designation | Source or reference | Identifiers | Additional information |
|---|---|---|---|---|
| Cell line (human) | U2OS cells | ATCC | HTB-96 | |
| Transfected construct (human) | HaloTag-Myo10-Flag (plasmid) | This paper | | See supplemental for DNA sequence |
| Antibody | Anti-Myosin10 antibody (rabbit polyclonal) | Novus | NBP1-87748 (RRID:AB_11035627) | 1:10,000 for western blot |
| Antibody | Anti-beta-tubulin antibody (mouse monoclonal) | Invitrogen | 22833 (RRID:AB_2533072) | 1:10,000 for western blot |
| Antibody | Anti-Rabbit-HRP antibody (goat polyclonal) | Cell Signaling | 7074 (RRID:AB_2099233) | 1:10,000 for western blot |
| Antibody | Anti-Mouse-HRP antibody (horse polyclonal) | Cell Signaling | 7076 (RRID:AB_330924) | 1:10,000 for western blot |
| Peptide, recombinant protein | Laminin | Sigma-Aldrich | CC095-M | |
| Peptide, recombinant protein | HaloTag standard protein | Promega | G4491 | |
| Commercial assay or kit | SuperSignal West Femto chemiluminescent substrate | Thermo Scientific | 34094 | |
| Other | Gibco 1x DMEM | Thermo Fisher | 11995073 | Cell media |
| Other | Accutase | Corning | MT25058CI | Enzyme cell detachment media |
| Other | FuGENE HD Transfection reagent | Promega | E2311 | Non-liposomal transfection reagent |
| Other | Lipofectamine 2000 | Invitrogen | 11668-027 | Cationic-lipid transfection reagent |
| Other | #1.5 coverglass bottom 35 mm Petri dishes | Cellvis, MatTek | D35-10-1.5-N, P35G-1.5-14-C | Coverglass for live-cell imaging |
| Other | Ibidi eight-well chamber slides | Ibidi | 80807 | Coverglass for fixed-cell imaging |
| Other | 4–20% Mini-PROTEAN TGX Stain-Free protein gel | Bio-Rad | 4568095 | Stain-free precast gels for SDS-PAGE |
| Other | TMR-HaloLigand | Promega | G8251 | Fluorophore-labeled HaloLigand used for visualizing Myo10 |
| Other | Alex Fluor 647 Phalloidin | Invitrogen | A22287 | 1 mM working concentration to label actin |
| Other | Bio-Beads SM-2 | Bio-Rad | 152-8920 | Resin to remove excess dye when testing HaloLigand labeling efficiency |

## Plasmids and constructs

Full-length human Myo10 sequence was constructed in a pTT5 vector backbone plasmid by *Gibson et al., 2009*. The construct includes an N-terminal Halotag (Promega; GenBank: JF920304.1), the full-length human Myo10 sequence (nucleotide sequence from GenBank: BC137168.1), and C-terminal Flag-tag (GATTATAAAGATGATGATGATAAA). A complete DNA sequence of the insert is included in supplemental information.

## Cell culture

U2OS cells (ATCC HTB-96, tested negative for mycoplasma by PCR and DAPI staining) were passaged every 2 days and used under passage number 10 after thawing. Cells were grown in Gibco 1x DMEM (Thermo Fisher 11995073) supplemented with 10% fetal bovine serum at 37°C and 5% $CO_2$.

## Transient transfection and coverslip seeding for fixed-cell imaging

Cells were transiently transfected with 1 µg of the HaloTag-Myo10-FlagTag plasmid and 0.2 µg of a calmodulin plasmid using FuGENE HD Transfection reagent (Promega E2311). Forty-eight hours after transfection, cells were detached using Accutase (Corning MT25058CI) and seeded onto ibidi

eight-well chamber slides (ibidi 80807) coated with 20 µg/ml laminin (Sigma-Aldrich CC095-M) for imaging and onto six-well dishes for SDS-PAGE analysis. After 3–4 hr, cells were collected for SDS-PAGE analysis at the same time as cells were fixed for microscopy. Three bioreplicates were conducted.

## Transient transfection and coverslip seeding for live-cell imaging

Cells were transiently transfected with 1 µg of the HaloTag-Myo10-FlagTag plasmid and 0.2 µg of a calmodulin plasmid using Lipofectamine 2000 (Invitrogen 11668-027). Twenty-four hours after transfection, cells were seeded onto #1.5 coverglass bottom 35 mm Petri dishes (Cellvis D35-10-1.5-N or MatTek P35G-1.5-14-C) coated with 20 µg/ml laminin for imaging and onto six-well dishes for SDS-PAGE analysis. After 3–4 hr, cells were collected for SDS-PAGE analysis right before cells were live-imaged. Three bioreplicates were conducted.

## Preparing cell lysates for SDS-PAGE

Cells growing in the six-well plate were first incubated with Accutase for 15 min at room temperature. For live-cell experiments, the Accutase additionally contained 0.75 µM TMR-HaloLigand (Promega G8251). Accutase was neutralized with DMEM+10% FBS and the cells were collected with a 3:30 min spin at 400 rcf at room temperature. The pellet was resuspended in 100 µl cell media, and 10 µl of the cells were removed and mixed with 1x Trypan blue for cell counting. The remaining cells were combined with 400 µl cell media and subjected to a 5 min, 400 rcf spin at room temperature. Cells were moved to ice post-spin and lysed in RIPA buffer containing 1 mM PMSF. Lysate samples were stored at –80°C.

## SDS gel analysis for Myo10 quantitation

HaloTag standard protein (Promega G4491) samples, with 0.5 µM TMR-HaloLigand, were prepared for final masses of 1.25 ng, 2.5 ng, 5 ng, 10 ng, and 15 ng and cell lysate samples (from live- and fixed-cell experiments) containing 50,000 cells on the gel. Excess TMR-HaloLigand was incubated for 10 min on ice with only lysates from the fixed-cell experiments. All samples were supplemented with an SDS loading buffer (0.1 M DTT, 2% SDS, 0.05% bromophenol blue, 0.05 M Tris-HCl, 10% glycerol, pH 6.8) and heated at 70°C for 10 min before loading onto a 4–20% Mini-PROTEAN TGX Stain-Free protein gel (Bio-Rad 4568095). The gel was run at 180 V for 45 min and imaged on a ChemiDoc. Images of the stain-free, AF647, and rhodamine channels were taken. We used the Gel Analysis plug-in in ImageJ to compare signal intensity of the gel bands. A standard curve was generated from the HaloTag standard protein TMR signal to estimate the number of Myo10 molecules per total transfected cells loaded (total cells loaded times the fraction transfected). Microscopy was used to count the percentage of transfected cells from ~105 to 190 randomly surveyed cells per bioreplicate.

## Western blotting for Myo10 in U2OS cells

We loaded 50,000 cells of wildtype and Myo10-transfected U2OS cells for SDS gel analysis as described earlier. The gel was transferred to PVDF membrane using a Pierce Power Blotter semi-dry electrophoretic transfer cell (1.3 A constant for 12 min). The blot was blocked with 5% milk in 1x TBST, 1 hr shaking at room temperature. For 1 hr at room temperature, the blot was incubated with anti-Myosin10 (Novus NBP1-87748) and anti-β-tubulin (Invitrogen 22833) antibodies in 5% milk in 1x TBST. The blot was rinsed with 5% milk in 1x TBST for 10 min, three total washes. Then, the blot was incubated with anti-Rabbit-HRP (Cell Signaling 7074) and anti-Mouse-HRP (Cell Signaling 7076) antibodies in 1x TBST, 1 hr shaking at room temperature. After three 10 min washes with 1x TBST, chemiluminescent substrate (Thermo Scientific, SuperSignal West Femto, 34094) was applied to the blot for 2 min and subsequently imaged on a Chemidoc.

## TMR-HaloLigand labeling efficiency

To test TMR-HaloLigand labeling efficiency, 6.8 µM of HaloTag standard protein was labeled with TMR-HaloLigand in 5.8× molar excess in a 20 µl reaction volume, on ice for 10 min. Bio-Beads SM-2 (Bio-Rad 152-8920) were washed with methanol 3×, distilled water 3×, and PBS 3× before they were added to ¼ volume of the reaction. After 1 hr and 40 min at room temperature in the dark, the reaction tube was spun for 3 min, 600 rcf spin. The supernatant was recovered and the absorbance spectrum was measured on Nanodrop. Absorbances at 280 nm and 553 nm were used for protein

and dye concentrations, respectively. To account for protein that may have nonspecifically bound to the beads, pre-bead TMR-HaloLigand-labeled HaloTag standard protein was loaded on an SDS-PAGE gel alongside the post-bead supernatant. Comparing the signal intensities of pre- and post-bead gel bands allowed for a corrected post-beads protein concentration value.

## Ligand labeling saturation for microscopy

To test ligand labeling saturation for microscopy, cells were fixed for 20 min in a solution comprising: 4% PFA, 0.08% Triton, 1:1000 DAPI in PEM buffer, and a series of TMR-HaloLigand concentrations (in µM: 0.05, 0.1, 0.5, 1, 2.5, 5). Live cells were incubated with DMEM+10% FBS containing TMR-HaloLigand (in µM: 0.05, 0.1, 0.5, 0.75, 1, 2.5 µM) at 37°C, 5% $CO_2$ in the dark. After 10 min and immediately prior to imaging, cells were washed and incubated with DMEM+10% FBS+GOC (an enzymatic oxygen-scavenger system: 4.5 mg/ml glucose, 0.5% β-mercaptoethanol, 4.3 mg/ml glucose oxidase, and 0.7 mg/ml catalase).

Image analysis was done using in-house Python scripts. Images were background-subtracted using a rolling ball radius of 50 pixels in ImageJ. To obtain total intracellular Myo10 signal in fixed cells, the phalloidin stain was used as a guide to manually draw bound cells. Filopodial puncta, segmented by watershed segmentation, were used to generate a convex hull mask. Signal inside the mask was summed.

To obtain total intracellular Myo10 signal in live cells, filopodial puncta, segmented by watershed segmentation, were used to generate a convex hull mask. Signal inside the convex hull mask was summed. Because there was higher, uneven background that persisted after the rolling ball subtraction in higher TMR-HaloLigand live samples, segmented Myo10 puncta were summed and plotted as well. Plotting only segmented puncta appeared to reduce the contribution of background pixels inside the convex hull mask that could falsely inflate final Myo10 intracellular signal.

## Fixed-cell fluorescence microscopy

For Myo10 fixed-cell imaging, cells were fixed for 20 min in a solution comprising: 4% PFA, 0.08% Triton, 2.5 µM TMR-HaloLigand, and 1:1000 DAPI in PEM buffer. After three PBS washes (4 min each), cells were stained with 1 mM Phalloidin-AF647 (Invitrogen A22287) in 1% BSA for 20 min. Cells were washed 3× with PBS (4 min each) before immediate imaging. Fluorescence images were taken on an Axiovert using a 60× water objective and a gain of 125 ms and 175 ms exposure (below detector saturation). To prevent photobleaching, cells were kept in the dark and imaged immediately upon illumination of the selected field of view. Samples were imaged within a day of preparation. TMR was visualized using green light and AF633 was visualized with red light.

## Analyzing fixed-cell images

Image analysis was done using in-house Python scripts. To calculate the number of Myo10 molecules in the fluorescence images, the Myo10-stained images were first background-subtracted: the average signal of a 56×56 pixel square near the cell body was calculated and subtracted from each pixel of the TIFF image. Next, the phalloidin-stained cell image and Myo10-stained image were filtered to remove non-cell objects and subjected to watershed segmentation. To generate a 'cell body mask', an erosion function followed by an opening function were applied to the phalloidin-stained cell image. The Myo10 image mask was defined as the 'total cell mask.' The 'cell body mask' was subtracted from the 'total cell mask' for a 'filopodia mask.' Connected component analysis (CCA) was performed on the 'filopodia mask' to obtain masks for all filopodia-localized Myo10 puncta. For the CCA, pixels were considered neighbors if they were connected through a maximum of two orthogonal hops. Segmented puncta (i.e. the connected regions) were then inspected in Napari (*Sofroniew et al., 2022*). Puncta belonging to the same filopodium were manually combined into the same filopodium mask.

To convert fluorescence intensities by microscopy into the number of Myo10 molecules, let $I_n$ represent the sum of the fluorescence signal for n =~50 cell images in a bioreplicate (each background-corrected, and each integrated over their 'total cell masks'). This fluorescence signal arises from n⟨m⟩ molecules, where ⟨m⟩ denotes the expected number of Myo10 molecules per transfected cell as determined by the SDS-PAGE analysis. Let $I_{ROI}$ represent the background-corrected and summed

intensity of a single region-of-interest (e.g. a filopodial punctum, or one cell). The number of Myo10 molecules found within that region of interest, r, is then:

$$r = n \langle m \rangle \frac{I_{ROI}}{I_n}$$

This procedure was repeated using the ⟨m⟩, n, and $I_n$ values for each bioreplicate (see *Figure 1—figure supplement 3* for a graphical representation of the math). To calculate percentage of Myo10-positive filopodia, total filopodia per cell were manually counted.

## Measuring Myo10 orientation distribution in fixed-cell filopodia

To generate the rose plots of the Myo10 distribution in filopodia, each cell was divided into 20 radial sections from the cell's center of mass. The Myo10 filopodial molecules were averaged within each section. The section with the highest molecules was aligned to degree 0 for each cell's rose plot. Total count of Myo10 puncta analyzed per section was also calculated. Puncta were not manually combined if belonging to the same filopodium. Therefore, puncta that were at the border of two radial sections were counted once for each section. For the rose plots in panel 2D, a randomly selected empty Myo10 section was aligned to degree 0 if more than one section contained no Myo10. The standard error bars represent the standard error of the mean molecules per section after 500 iterations of bootstrapping. In each iteration, 150 cells were randomly selected with replacement.

## Measuring local Myo10 concentrations at the tips of fixed-cell filopodia

To obtain estimates of local Myo10 concentrations in filopodia tips, 10 Myo10 tip-localized puncta were randomly chosen from three different cell images of each bioreplicate set. The length of each punctum was measured in ImageJ. The volume occupied by the Myo10 punctum was estimated using the volume of a cylinder: length = height and width = 2*radius, where radius was assumed to be 100 nm (reported average radius) because of the resolution limit. The signal intensity of the Myo10 punctum was converted to molecules as described above.

Available actin monomer concentration for Myo10 binding is described in *Figure 3—figure supplement 2*. Available membrane surface for Myo10 binding is described in *Figure 3—figure supplement 3*.

## Staining cells for live-cell imaging

For Myo10 live imaging, cells were incubated with DMEM+10% FBS containing 0.75 µM TMR-HaloLigand at 37°C in the dark. After 10 min, cells were quickly washed 2× with DMEM+10% FBS washes, and one 5 min wash. Immediately before imaging, cells were replaced with DMEM+10% FBS containing 1× GOC. Live cells were imaged on a 37°C heated objective using a gain of 100 ms and 300 ms exposure (below detector saturation). Ten random cells per bioreplicate were recorded for <6 min (three bioreplicates in total). For each bioreplicate, additional snapshots of ~50 living cells were taken to sample the intensity distribution for the fluorescence intensity-to-molecule conversions. Fixed-cell samples were prepared in parallel to obtain transfection efficiencies of the bioreplicates.

## Analyzing live-cell movies

Image analysis was partially done using in-house Python scripts. To get an intensity distribution for the fluorescence intensity-to-molecule conversions, the Myo10-stained snapshots were first background-subtracted using a rolling ball radius of 50 pixels in ImageJ. To obtain total intracellular Myo10 signal, filopodial puncta were used as bounding edges for a manually drawn mask. Signal inside the drawn mask was summed. Conversion of fluorescence signal intensity to number of molecules was performed as described above for fixed cells. Myo10 puncta in the live cell movies were segmented and tracked using the Fiji plug-in TrackMate (*Tinevez et al., 2017*). Trackmate parameters used: LoG detector with estimated blob diameter = 1 µm, median filter, sub-pixel localization; LAP tracker with frame to frame linking = 0.5 µm, track segment gap closing = 1 µm (max frame gap = 2), track segment splitting = 0.5 µm, track segment merging = 1 µm.

After completing TrackMate analysis, all trajectories in the cell movies were manually inspected for occurrence of three types of filopodial events. The start of the three events are defined by when TrackMate first segments the punctum. 'Initiation' is when a Myo10 punctum gathers at the plasma

membrane and then shoots into a filopodium. 'Second-phase elongation' is a new Myo10 punctum that emerges from an existing one. 'Retraction' is when a Myo10 punctum, observed in frame 1 of the movie, begins moving back toward the cell body. The track IDs were recorded for each trajectory type and further inspected. A few puncta that TrackMate segmented were manually removed because they were clearly background noise or incorrectly tracked puncta (e.g. a punctum from another filopodium that briefly crosses paths, or filopodial puncta that were detected by TrackMate only well after initiation). Any selected trajectories containing Myo10 puncta merging or splitting events were excluded from velocity analysis.

For *Figure 4* parts A–C, Myo10 puncta molecules in the first two frames of each trajectory were averaged. For *Figure 4* parts D–F, the starting punctum molecules are subtracted from subsequent frames of each trajectory. Parts D–E follow the trajectory of each punctum that is initially identified for a certain filopodial event, but then those puncta could have engaged in other activities during their lifetime (e.g. retract). GAM fitting was only applied to a subset of the total data to prevent a few long trajectories from dominating the trend. For *Figure 4* parts G–H, note that velocity analysis includes a few Myo10 puncta that switch direction within a single trajectory (e.g. a retracting punctum that then elongates). For *Figure 4* parts G–I, puncta merging or splitting events were excluded from analysis.

## Statistical analysis

Statistical tests were performed using R. The statistical tests used for each quantification are stated in the figure legends and within the main text when each figure is initially referenced.

## Acknowledgements

We thank Eugene Schauchuk for designing and cloning the Myo10 construct, and Prof. David Kovar, Department of Molecular Genetics and Cell Biology, University of Chicago, for critical comments on the manuscript. This work was supported by the University of Chicago MCB Training Grant (T32 GM144292) and the NSF Graduate Research Fellowship (2140001) (to JS) and NIH grants R01GM124272 and R01GM149073 (to RSR).

## Additional information

### Competing interests

Ronald S Rock: RSR is a consultant for Cyntegron Therapeutics. The other author declares that no competing interests exist.

### Funding

| Funder | Grant reference number | Author |
|---|---|---|
| National Institute of General Medical Sciences | T32 GM144292 | Julia Shangguan |
| National Science Foundation | 2140001 | Julia Shangguan |
| National Institute of General Medical Sciences | R01 GM124272 | Ronald S Rock |
| National Institute of General Medical Sciences | R01 GM149073 | Ronald S Rock |

The funders had no role in study design, data collection and interpretation, or the decision to submit the work for publication.

### Author contributions

Julia Shangguan, Conceptualization, Data curation, Software, Formal analysis, Funding acquisition, Investigation, Visualization, Methodology, Writing - original draft, Writing – review and editing; Ronald S Rock, Conceptualization, Resources, Data curation, Formal analysis, Supervision, Funding acquisition, Validation, Methodology, Project administration, Writing – review and editing

## Author ORCIDs
Julia Shangguan http://orcid.org/0000-0002-6293-1519
Ronald S Rock https://orcid.org/0000-0003-2188-7272

Joint Public Review: https://doi.org/10.7554/eLife.90603.4.sa1
Author response https://doi.org/10.7554/eLife.90603.4.sa2

## Additional files

### Supplementary files
- MDAR checklist
- Source code 1. A descriptive text file explaining each subfolder is included within the zip file.

### Data availability
All materials generated in this study are available from the corresponding author under a materials transfer agreement with the University of Chicago. Data and code used for data analysis can be found in the supplementary materials for this article.

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
