## [Editor Report · eLife assessment]

The manuscript proposes an alternative method by SDS-PAGE calibration of Halo-Myo10 signals to quantify myosin molecules in filopodia and discusses different scenarios regarding myosin 10 working models to explain intracellular diffusion and targeting to filopodia. Overall, the paper is elegantly written and the methodology is **valuable** in its descriptive potential as these are key numbers to know to ultimately decipher the cellular mechanism of Myo10 action as well as understand the molecular composition of a Myo10-generated filopodium. The evidence for the conclusions is **compelling**, but there are limitations to this study which should be kept in mind when applying this method to other systems.

---

## [Referee Report · Joint Public Review]

The paper sought to determine the number of myosin 10 molecules per cell and localized to filopodia, where they are known to be involved in formation, transport within, and dynamics of these important actin-based protrusions. The authors used a novel method to determine the number of molecules per cell. First, they expressed HALO tagged Myo10 in U20S cells and generated cell lysates of a certain number of cells and detected Myo10 after SDS-PAGE, with fluorescence and a stained free method. They used a purified HALO tagged standard protein to generate a standard curve which allowed for determining Myo10 concentration in cell lysates and thus an estimate of the number of Myo10 molecules per cell. They also examined the fluorescence intensity in fixed cell images to determine the average fluorescence intensity per Myo10 molecule, which allowed the number of Myo10 molecules per region of the cell to be determined. They found a relatively small fraction of Myo10 (6%) localizes to filopodia. There are hundreds of Myo10 in each filopodia, which suggests some filopodia have more Myo10 than actin binding sites. Thus, there may be crowding of Myo10 at the tips, which could impact transport, the morphology at the tips, and dynamics of the protrusions themselves. Overall, the study forms the basis for a novel technique to estimate the number of molecules per cell and their localization to actin-based structures. The implications are broad also for being able to understand the role of myosins in actin protrusions, which is important for cancer metastasis and wound healing.

Comments on latest version (from the Reviewing Editor):

One of the main critiques that still remains is that the results were derived from experiments with overexpressed Myo10 and therefore are hard to extrapolate to physiological conditions. Measurement were also only performed in a single cell line. The authors counter this critique with the argument that their results provide insight into a system in which Myo10 is a limiting factor for controlling filopodia formation. They demonstrate that U20S cells do not express detectable levels of Myo10 and thus introducing Myo10 expression demonstrates how triggering Myo10 expression impacts filopodia. An example is given of how melanoma cells often heavily upregulate Myo10.

---

## [Author Response]

The following is the authors’ response to the previous reviews.

**Public Reviews:**

**Reviewer #1 (Public Review):**
Summary:The manuscript proposes an alternative method by SDS-PAGE calibration of Halo-Myo10 signals to quantify myosin molecules at specific subcellular locations, in this specific case filopodia, in epifluorescence datasets compared to the more laborious and troublesome single molecule approaches. Based on these preliminary estimates, the authors developed further their analysis and discussed different scenarios regarding myosin 10 working models to explain intracellular diffusion and targeting to filopodia.Strengths:I confirm my previous assessment. Overall, the paper is elegantly written and the data analysis is appropriately presented. Moreover, the novel experimental approach offers advantages to labs with limited access to high-end microscopy setups (super-resolution and/or EM in particular), and the authors proved its applicability to both fixed and live samples.Weaknesses:Myself and the other two reviewers pointed to the same weakness, the use of protein overexpression in U2OS. The authors claim that Myosin10 is not expressed by U2OS, based on Western blot analysis. Does this completely rule out the possibility that what they observed (the polarity of filopodia and the bulge accumulation of Myo10) could be an artefact of overexpression? I am afraid this still remains the main weakness of the paper, despite being properly acknowledged in the Limitations.

Respectfully, our observations do not capture an “artefact” of overexpression but rather the “response” to overexpression. Our goal in this project was to overexpress Myo10 in a situation where it is the limiting reagent for generating filopodia. As Reviewer 3 notes below, overexpression shows that filopodial tips “can accommodate a surprisingly (shockingly) large number of motors.” This is exactly the point. Reviewer 2 considered our handling of this issue to be a strength of the paper. As far as whether bulges occur in endogenous Myo10 systems, please see our comments to Reviewer 3.

I consider all the remaining issues I expressed during the first revision solved.
**Reviewer #2 (Public Review):**
Summary:The paper sought to determine the number of myosin 10 molecules per cell and localized to filopodia, where they are known to be involved in formation, transport within, and dynamics of these important actin-based protrusions. The authors used a novel method to determine the number of molecules per cell. First, they expressed HALO tagged Myo10 in U20S cells and generated cell lysates of a certain number of cells and detected Myo10 after SDS-PAGE, with fluorescence and a stained free method. They used a purified HALO tagged standard protein to generate a standard curve which allowed for determining Myo10 concentration in cell lysates and thus an estimate of the number of Myo10 molecules per cell. They also examined the fluorescence intensity in fixed cell images to determine the average fluorescence intensity per Myo10 molecule, which allowed the number of Myo10 molecules per region of the cell to be determined. They found a relatively small fraction of Myo10 (6%) localizes to filopodia. There are hundreds of Myo10 in each filopodia, which suggests some filopodia have more Myo10 than actin binding sites. Thus, there may be crowding of Myo10 at the tips, which could impact transport, the morphology at the tips, and dynamics of the protrusions themselves. Overall, the study forms the basis for a novel technique to estimate the number of molecules per cell and their localization to actin-based structures. The implications are broad also for being able to understand the role of myosins in actin protrusions, which is important for cancer metastasis and wound healing.Strengths:The paper addresses an important fundamental biological question about how many molecular motors are localized to a specific cellular compartment and how that may relate to other aspects of the compartment such as the actin cytoskeleton and the membrane. The paper demonstrates a method of estimating the number of myosin molecules per cell using the fluorescently labeled HALO tag and SDS-PAGE analysis. There are several important conclusions from this work in that it estimates the number of Myo10 molecules localized to different regions of the filopodia and the minimum number required for filopodia formation. The authors also establish a correlation between number of Myo10 molecules filopodia localized and the number of filopodia in the cell. There is only a small % of Myo10 that tip localized relative to the total amount in the cell, suggesting Myo10 have to be activated to enter the filopodia compartment. The localization of Myo10 is log-normal, which suggests a clustering of Myo10 is a feature of this motor.One of the main critiques of the manuscript was that the results were derived from experiments with overexpressed Myo10 and therefore are hard to extrapolate to physiological conditions. The authors counter this critique with the argument that their results provide insight into a system in which Myo10 is a limiting factor for controlling filopodia formation. They demonstrate that U20S cells do not express detectable levels of Myo10 (supplementary Figure 1E) and thus introducing Myo10 expression demonstrates how triggering Myo10 expression impacts filopodia. An example is given how melanoma cells often heavily upregulate Myo10.In addition, the revised manuscript addresses the concerns about the method to quantitate the number of Myo10 molecules per cell and therefore puncta in the cell. The authors have now made a good faith effort to correct for incomplete labeling of the HALO tag (Figure 2A-C, supplementary Figure 2D-E). The authors also address the concerns about variability in transfection efficiency (Figure 1D-E).A very interesting addition to the revised manuscript was the quantitation of the number of Myo10 molecules present during an initiation event when a newly formed filopodia just starts to elongate from the plasma membrane. They conclude that 100s of Myo10 molecules are present during an initiation event. They also examined other live cell imaging events in which growth occurs from a stable filopodia tip and correlated with elongation rates.Weaknesses:The authors acknowledge that a limitation of the study is that all of the experiments were performed with overexpressed Myo10. They address this limitation in the discussion but also provide important comparisons for how their work relates to physiological conditions, such as melanoma cells that only express large amounts of Myo10 when they are metastatic. Also, the speculation about how fascin can outcompete Myo10 should include a mechanism for how the physiological levels of fascin can complete with the overabundance of Myo10 (page 10, lines 401-408).

We have expanded the discussion about fascin competing with high concentrations of Myo10 in filopodial tips on pg. 15. The key feature is that fascin binding in a bundle is essentially irreversible, so it wins if any space opens up and it manages to bind before the next Myo10 arrives.

**Reviewer #3 (Public Review):**
SummaryThe work represents progress in quantifying the number of Myo10 molecules present in the filopodia tip. It reveals that cells overexpressing fluorescently labeled Myo10 that the tip can accommodate a wide range of Myo10 motors, up to hundreds of molecules per tip.The revised, expanded manuscript addresses all of this reviewer's original comments. The new data, analysis and writing strengthen the paper. Given the importance of filopodia in many cellular/developmental processes and the pivotal, as yet not fully understood role of Myo10 in their formation and extension, this work provides a new look at the nature of the filopodial tip and its ability to accommodate a large number of Myo10 motor proteins through interactions with the actin core and surrounding membrane.Specific comments -(1) One of the comments on the original work was that the analysis here is done using cells ectopically expressing HaloTag-Myo10. The author's response is that cells express a range of Myo10 levels and some metastatic cancer cells, such as breast cancer, have significantly increased levels of Myo10 compared to non-transformed cell lines. It is not really clear how much excess Myo10 is present in those cells compared to what is seen here for ectopic expression in U2OS cells, making a direct correspondence difficult.

We agree, a direct correspondence is difficult, and is further complicated by other variables (e.g., expression levels of Myo10 activators, cargoes, fascin, or other filopodial components) that may differ among cell lines. Properly sorting this out will require additional work in a few key cellular systems.

However, there are two points to keep in mind that somewhat mitigate this concern. First, because ectopic expression of Myo10 causes an ~30x increase in the number of filopodia, the activated Myo10 population is divided over that larger filopodial population. Second, the log-normal distribution of Myo10 across filopodia has a long tail, which means that some cells with low levels of Myo10 will concentrate that Myo10 in a few filopodia.

In response to comments about the bulbous nature of many filopodia tips the authors point out that similar-looking tips are seen when cells are immunostained for Myo10, citing Berg & Cheney (2002). In looking at those images as well as images from papers examining Myo10 immunostaining in metastatic cancer cells (Arjonen et al, 2014, JCI; Summerbell et al, 2020, Sci Adv) the majority of the filopodia tips appear almost uniformly dot-like or circular. There is not too much evidence of the elongated, bulbous filopodial tips seen here.

Yes, the tips in Berg and Cheney are circular, but their size varies considerably (just as a balloon is roughly circular, its size varies with the amount of air it contains). Non-bulbous filopodial tips have a theoretical radius of ~100 nm, which is below the diffraction limit. However, many of the filopodial tips are larger than the diffraction limit in Berg and Cheney, Fig. 1a. We cropped and zoomed in the images to show each fully visible filopodial tip

We attempted to perform a similar analysis of the images in Arjonen and Summerbell. Unfortunately, their images are too small to do so.

However, in reconsidering the approach and results, it is the case that the finding here do establish the plasticity of filopodia tips that can accommodate a surprisingly (shockingly) large number of motors. The authors discuss that their results show that targeting molecules to the filopodia tip is a relatively permissive process (lines 262 - 274). That could be an important property that cells might be able to use to their advantage in certain contexts.(2) The method for arriving at the intensity of an individual filopodium puncta (starting on line 532 and provided in the Response), and how this is corrected for transfection efficiency and the cell-to-cell variation in expression level is still not clear to this reviewer. The first part of the description makes sense - the authors obtain total molecules/cell based on the estimation on SDS-PAGE using the signal from bound Halo ligand. It then seems that the total fluorescence intensity of each expressing cell analyzed is measured, then summed to get the average intensity/cell. The 'total pool' is then arrived at by multiplying the number of molecules/cell (from SDS-PAGE) by the total number of cells analyzed. After that, then: 'to get the number of molecules within a Myo10 filopodium, the filopodium intensity was divided by the bioreplicate signal intensity and multiplied by 'total pool.' ' The meaning of this may seem simple or straightforward to the authors, but it's a bit confusing to understand what the 'bioreplicate signal intensity' is and then why it would be multiplied by the 'total pool'. This part is rather puzzling at first read.

We agree, such information is critical. We have now revised this description with more precise terms and have included a formula on pg. 20.

Since the approach described here leads the authors to their numerical estimates every effort should be made to have it be readily understood by all readers. A flow chart or diagram might be helpful.

We have added a diagram of the calculations to the supplemental material (Figure 1—figure supplement 3). We hope that both changes will make it easier for others to follow our work.

(3) The distribution of Myo10 punctae around the cell are analyzed (Fig 2E, F) and the authors state that they detect 'periodic stretches of higher Myo10 density along the plasma membrane' (line 123) and also that there is correlation and anti-correlation of molecules and punctae at opposite ends of the cells.In the first case, it is hard to know what the authors really mean by the phrase 'periodic stretches'. It's not easy to see a periodicity in the distribution of the punctae in the many cells shown in Supp Fig 3. Also, the correlation/anti-correlation is not so easily seen in the quantification shown in Fig 2F. Can the authors provide some support or clarification for what they are stating?

The periodic pattern that we refer to is most apparent in the middle panels of Fig. 2E, F. These panels show the density of Myo10 puncta. These puncta numbers closely correspond to filopodia counts, with the caveat that some filopodia might have multiple puncta. This periodic density might not be as apparent in the raw data shown in Supp. Fig. 3. We have therefore rewritten this paragraph to clarify our observations (pg. 6).

(4) The authors are no doubt aware that a paper from the Tyska lab that employs a completely different method of counting molecules arrives at a much lower number of Myo10 molecules at the filopodial tip than is reported here was just posted (Fitz & Tyska, 2024, bioRxiv, DOI: 10.1101/2024.05.14.593924).While it is not absolutely necessary for the authors to provide a detailed discussion of this new work given the timing, they may wish to consider adding a note briefly addressing it.

We are aware of this manuscript and that it uses a different approach for calibrating the fluorescence signal in microscopy. However, we are not comfortable commenting on that manuscript at this time, given that it has not yet been peer reviewed with the chance for author revisions.

**Recommendations for the authors:**

**Reviewer #1 (Recommendations For The Authors):**
The manuscript the authors are now presenting does not comply with the formatting limits of a Short report, but it is instead presented as a full article type. I believe the authors could shorten the Discussion, and meet the criteria for a more appropriate Short Report format.For instance, I continue to believe that the study of truncation variants could sustain the claim that membrane binding represents the driving force that leads to Myo10 accumulation. I understand the authors want to address these mechanisms in a follow-up story, for this reason, I encourage them to shorten the discussion, which seems unnecessarily long for a technique-based manuscript.

In the first round of review, Reviewer 3 asked us to expand the discussion. Given that, we are happy with where we have landed on the length of the discussion.

Figure 2, could include some images to facilitate the readers on the different messages of the two rose plots E and F, by picking one of the examples from the supplementary Figure 3

We have now added a supplemental figure showing an example cell (Fig. 2 figure supplement 2). But please note that the averaging of ~150 cells (Fig. 2E, F) should be more reliable to show these overall trends.

**Reviewer #2 (Recommendations For The Authors):**
Also, the speculation about how fascin can outcompete Myo10 should include a mechanism for how the physiological levels of fascin can complete with the overabundance of Myo10 (page 10, lines 401-408).

As noted above, we have now clarified this point.

**Reviewer #3 (Recommendations For The Authors):**
line 495 - what is GOC?

We have now defined this oxygen scavenger system in the main text.

lines 603/604 - it is stated that 'velocity analysis does not only account for Myo10 punctum that moved away from the starting point of the trajectory.' It's not clear what this really means.

The sentence now reads: "For Figure 4 parts G-H, note that velocity analysis includes a few Myo10 puncta that switch direction within a single trajectory (e.g., a retracting punctum that then elongates)."

References #4 and #14 are the same.

Thank you for catching that; it has now been corrected.

Fig 1C - the plot for signal intensity versus fmol of protein has numbers for the standard and then live and fixed cells. While the R2 value is quite good, it seems a bit odd that the three (?) data points for live cells are all quite small relative to the fixed cells and all bunched together at the left side of the plot.

As mentioned in the main text, the time post-transfection has a noticeable effect on the level of Myo10 expression. The three fixed-cell bioreplicates had higher Myo10 expression because they were analyzed 48 hours post-transfection compared to the three live-cell bioreplicates (24 hours). Therefore, the fixed cell data points are larger in value because they represent more molecules, and the live cell data points are on the left side of the plot because they represent fewer molecules.